# Spatial analysis of COVID-19 incidence and the sociodemographic context in Brazil

**Carlos Eduardo Raymundo**[1]*, **Marcella Cini Oliveira**[2], **Tatiana de Araujo Eleuterio**[1,3], **Suzana Rosa André**[4], **Marcele Gonçalves da Silva**[2], **Eny Regina da Silva Queiroz**[1], **Roberto de Andrade Medronho**[1,2]

1 Instituto de Estudos em Saúde Coletiva, Universidade Federal do Rio de Janeiro, Rio de Janeiro, State of Rio de Janeiro, Brazil, 2 Faculdade de Medicina, Universidade Federal do Rio de Janeiro, Rio de Janeiro, State of Rio de Janeiro, Brazil, 3 Departamento de Enfermagem em Saúde Pública, Universidade do Estado do Rio de Janeiro, Rio de Janeiro, State of Rio de Janeiro, Brazil, 4 Escola de Enfermagem Anna Nery, Universidade Federal do Rio de Janeiro, Rio de Janeiro, State of Rio de Janeiro, Brazil

* caducer@gmail.com

**Data Availability Statement:** All relevant data are within the manuscript.

**Funding:** The author(s) received no specific funding for this work.

## Abstract

### Background

Identified in December 2019 in the city of Wuhan, China, the outbreak of COVID-19 spread throughout the world and its impacts affect different populations differently, where countries with high levels of social and economic inequality such as Brazil gain prominence, for understanding of the vulnerability factors associated with the disease. Given this scenario, in the absence of a vaccine or safe and effective antiviral treatment for COVID-19, nonpharmacological measures are essential for prevention and control of the disease. However, many of these measures are not feasible for millions of individuals who live in territories with increased social vulnerability. The study aims to analyze the spatial distribution of COVID-19 incidence in Brazil's municipalities (counties) and investigate its association with sociodemographic determinants to better understand the social context and the epidemic's spread in the country.

### Methods

This is an analytical ecological study using data from various sources. The study period was February 25 to September 26, 2020. Data analysis used global regression models: ordinary least squares (OLS), spatial autoregressive model (SAR), and conditional autoregressive model (CAR) and the local regression model called multiscale geographically weighted regression (MGWR).

### Findings

The higher the GINI index, the higher the incidence of the disease at the municipal level. Likewise, the higher the nurse ratio per 1,000 inhabitants in the municipalities, the higher the COVID-19 incidence. Meanwhile, the proportional mortality ratio was inversely associated with incidence of the disease.

**Competing interests:** The authors have declared that no competing interests exist.

## Discussion

Social inequality increased the risk of COVID-19 in the municipalities. Better social development of the municipalities was associated with lower risk of the disease. Greater access to health services improved the diagnosis and notification of the disease and was associated with more cases in the municipalities. Despite universal susceptibility to COVID-19, populations with increased social vulnerability were more exposed to risk of the illness.

## Introduction

Cases of pneumonia of unknown etiology were reported in December 2019 in the province of Hubei, China. A novel betacoronavirus was identified, similar to severe acute respiratory syndrome coronavirus (SARS-CoV) and Middle East respiratory syndrome coronavirus (MERS-CoV). The novel coronavirus is called severe acute respiratory syndrome coronavirus 2 (SARS-CoV-2). SARS-CoV-2 infection causes coronavirus disease 2019 (COVID-19), which can either course asymptomatically, as a flu-like syndrome, or evolve to acute respiratory distress syndrome (ARDS). On January 30, 2020, the World Health Organization (WHO) declared the disease a public health emergency of international concern. On March 11, 2020, the WHO declared COVID-19 a pandemic [1–4]. The first case in Brazil was reported on February 25, 2020. On March 20, the Brazilian Ministry of Health confirmed community transmission of SARS-CoV-2 throughout the country's territory and adopted mitigation measures since then to control the pandemic [5].

As of September 15, 2020, the WHO had recorded 29,155,581 confirmed cases of COVID-19, with 926,544 deaths [6]. Brazil is currently the world's third leading country in number of COVID-19 cases, with 4,345,610, and second in the number of deaths, with 132,006, for a case-fatality rate of 3.0% [7].

Since the vaccines are still in the experimental phase and there is no scientific evidence that corroborates the efficacy and safety of antiviral drugs in COVID-19, nonpharmacological interventions are priorities for reducing the number of cases, thereby avoiding overload of health services. Promotion of social distancing, restricting circulation of persons, wearing masks, and spreading information on measures in personal hygiene and prevention have been identified as the main strategies for fighting the disease [8]. Still, many of these strategies are not feasible for millions of individuals who live in irregular housing settlements, in territories characterized by increased social vulnerability, with precarious housing and sanitation.

Recently published Brazilian and international studies have found relations between sociodemographic, environmental, and healthcare factors and COVID-19 incidence, where structural conditions contribute to exposure to risk and the capacity for the community's recovery from the pandemic [9–20]. A study conducted in the Ceará state (Brazil), investigated the correlation of the human development index by municipalities with the incidence of COVID-19 [13]. A study carried out in the United States used the Gini index as a measure of income inequality [19]. Variables such as mean income, health care facilities, education level and race have also been included as potential risk factors for the disease [10, 11, 14–16, 20]. Nevertheless, the association of these factors with COVID-19 remains to be better understood.

Brazil is a country of continental proportions and major social and economic heterogeneity, and like many other emerging countries, it presents great potential for spread of the disease.

Knowledge of disease's spatial dynamic and its relations with social determinants is essential for the identification of areas with increased potential for spread of the infection,

prioritization of prevention and control measures in these areas, implementation of more restrictive social distancing, and the health system's preparation for treating cases.

In this sense, the use of tools to understand socioeconomic factors and levels of inequality associated with the development of the disease are relevant, such as the Gini coefficient, which is usually used to measure income inequality, being always non-negative with values between zero and one [21, 22].

Geographic information system (GIS) has been used to assess the spatial distribution of infectious diseases. In Brazil, the COVID-19 Panel presents updated data described in graphs, tables and maps. This data map can aid in analyzing the spread of COVID-19 and improving the quality of care. The characterization of risk areas could contribute to stakeholders decision-making during the pandemic. Thus, statistical techniques for spatial analysis to help determine relationship between several explanatory variables and disease outbreak.

This study aimed to analyze spatial distribution of COVID-19 incidence in Brazilian municipalities and to investigate the association between incidence of the disease and sociodemographic determinants to better understand the social context and the epidemic's spread in the country.

## Materials and methods

### Design

This analytical ecological study evaluated the association between demographic, socioeconomic, and healthcare covariables and COVID-19 incidence. The analytical units were Brazil's 5,570 municipalities. Brazil consists of 26 states and the Federal District. The states are grouped administratively into five major geographic regions (Fig 1).

### Data collection

The numbers of confirmed COVID-19 cases by municipality were obtained from the Coronavirus Panel, updated daily by the Ministry of Health (https://covid.saude.gov.br/). The study period was February 25 to September 26, 2020.

| State Code | State Name |
|---|---|
| RO | Rondônia |
| AC | Acre |
| AM | Amazonas |
| RR | Roraima |
| PA | Pará |
| AP | Amapá |
| TO | Tocantins |
| MA | Maranhão |
| PI | Piauí |
| CE | Ceará |
| RN | Rio Grande do Norte |
| PB | Paraíba |
| PE | Pernambuco |
| AL | Alagoas |
| SE | Sergipe |
| BA | Bahia |
| MG | Minas Gerais |
| ES | Espírito Santo |
| RJ | Rio de Janeiro |
| SP | São Paulo |

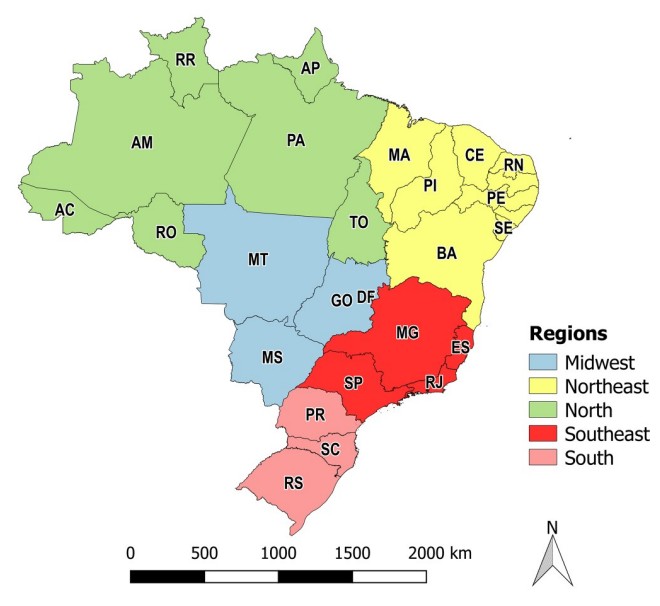

**Fig 1. Administrative division of Brazil.**

COVID-19 incidence by municipality (outcome) was calculated as the ratio between the absolute number of cases and the resident population in the municipality, multiplied by $10^4$. Data on the resident population by municipality correspond to the estimates by the Federal Accounts Court (TCU) for the year 2019 (http://tabnet.datasus.gov.br/cgi/deftohtm.exe?popsvs/cnv/popbr.def), based on census data for 2010 by the Brazilian Institute of Geography and Statistics (IBGE).

The study used demographic, socioeconomic, and healthcare covariables to investigate possible associations with the outcome. These variables were selected due to their public availability and relevance as social determinants of health. In the particular case of health determinants, it is important to highlight the relationship between variables of availability of health professionals (such as doctors and nurses) and socioeconomic development, since locations with greater development in general also have a more developed job markets, which, in turn, attracts such professionals. In this sense, the availability of these professionals is a proxy for social inequality [23]. Table 1 shows the covariables.

Table 1. Type and description of the study variables, data sources, and reference period.

| Type | Description of variable | Acronym | Data source | Period |
|---|---|---|---|---|
| Demographic | Estimated total population of municipality | TOTALPOP | DATASUS—Resident population—estimates from TCU | 2019 |
| | Proportion of elderly in municipality | ELDER | DATASUS–Resident population—estimates from TCU | 2019 |
| | Proportion of black or brown population in the municipality | BLKBRN | DATASUS—Resident population—estimates from TCU | 2019 |
| | Population density | POP_DENS | Brazilian Institute of Geography and Statistics—Population Census | 2019 |
| Healthcare | Physicians with employment contract per municipality per 1,000 inhabitants | DOCTRS | Ministry of Health—National Registry of Healthcare Establishments–CNES | January to June 2020 |
| | Nurses with employment contract per municipality per 1,000 inhabitants | NURS | Ministry of Health—National Registry of Healthcare Establishments–CNES | January to June 2020 |
| | Number of state primary outpatient clinics per municipality | ST_PRI_CLIN | Ministry of Health—National Registry of Healthcare Establishments–CNES | June 2020 |
| | Number of municipal primary outpatient clinics per municipality | MUN_PRI_CLIN | Ministry of Health—National Registry of Healthcare Establishments–CNES | June 2020 |
| | Number of state secondary outpatient clinics per municipality | ST_SEC_CLIN | Ministry of Health—National Registry of Healthcare Establishments—CNES | June 2020 |
| | Number of municipal secondary outpatient clinics per municipality | MUN_SEC_CLIN | Ministry of Health—National Registry of Healthcare Establishments—CNES | June 2020 |
| | Number of state tertiary outpatient clinics per municipality | ST_TER_CLIN | Ministry of Health—National Registry of Healthcare Establishments—CNES | June 2020 |
| | Number of municipal tertiary outpatient clinics per municipality | MUN_TER_CLIN | Ministry of Health—National Registry of Healthcare Establishments—CNES | June 2020 |
| | Number of state secondary hospitals per municipality | ST_SEC_HOSP | Ministry of Health—National Registry of Healthcare Establishments—CNES | June 2020 |
| | Number of municipal secondary hospitals per municipality | MUN_SEC_HOSP | Ministry of Health—National Registry of Healthcare Establishments—CNES | June 2020 |
| | Number of state tertiary hospitals per municipality | ST_TER_HOSP | Ministry of Health—National Registry of Healthcare Establishments—CNES | June 2020 |
| | Number of municipal tertiary hospitals per municipality | MUN_TER_HOSP | Ministry of Health—National Registry of Healthcare Establishments—CNES | June 2020 |
| | Total number of establishments (outpatient clinics and hospitals) | ESTABS | Ministry of Health—National Registry of Healthcare Establishments—CNES | June 2020 |

*(Continued)*

**Table 1.** (Continued)

| Type | Description of variable | Acronym | Data source | Period |
|------|------------------------|---------|-------------|--------|
| | Hospitalizations for pneumonia | PNEUMO | Ministry of Health–Hospital Information System—SIH | January 2019 to May 2020 |
| | Population covered by private health plans | PPLAN | National Supplementary Health Agency (ANS) | March 2020 |
| Socioeconomic | Log of mean monthly nominal household income | LOGINCOM | Brazilian Institute of Geography and Statistics—Population Census | 2010 |
| | GINI index | GINI | Brazilian Institute of Geography and Statistics—Population Census | 2010 |
| | Unemployment rate | UNEMPL_RT | General Registry of Employed and Unemployed (CAGED) | January to December 2019 |
| | Proportional mortality ratio or Swaroop and Uemura index | PMR | Mortality Information System (SIM) | 2018 |
| | Municipal human development index–MHDI | MHDI | Human Development Atlas | 2010 |
| | Municipal human development index–MHDI Income | MHDI_INCOM | Human Development Atlas | 2010 |
| | Municipal human development index–MHDI Life expectancy | MHDI_LONG | Human Development Atlas | 2010 |
| | Municipal human development index–MHDI Education | MHDI_EDU | Human Development Atlas | 2010 |
| | Proportion of population 15 years or older with 0 to 4 years of schooling | EDUC0TO4 | Brazilian Institute of Geography and Statistics—Population Census | 2010 |
| | Proportion or population 15 years or older with 5 to 8 years of schooling | EDUC5TO8 | Brazilian Institute of Geography and Statistics—Population Census | 2010 |
| | Proportion of population 15 years or older with 9 or more years of schooling | EDUC9 | Brazilian Institute of Geography and Statistics—Population Census | 2010 |
| | Proportion of households with running water | WATER | Brazilian Institute of Geography and Statistics—Population Census | 2010 |
| | Proportion of households with public sewage disposal | SEWAGE | Brazilian Institute of Geography and Statistics—Population Census | 2010 |
| | Proportion of households with public garbage collection | GRBG | Brazilian Institute of Geography and Statistics—Population Census | 2010 |
| | Percentual of population with monthly income 1 to 4 minimum wages | INCOM_1TO4 | Brazilian Institute of Geography and Statistics—Population Census | 2010 |
| | Percentual of population with monthly income 1 to 2 minimum wages | INCOM_2TO5 | Brazilian Institute of Geography and Statistics—Population Census | 2010 |

## Data analysis

Local empirical Bayesian smoothing was used to reduce the effect of the instability in COVID-19 incidence in Brazilian municipalities, weighting the incidence of the disease in a municipality with the incidence in the neighboring municipalities. Besides, since the incidence rates' distribution was not Gaussian, log transformation was applied to approach it to normal distribution. The study's outcome was thus the log COVID-19 incidence with Bayesian smoothing (LOGCOV), hereinafter "incidence of the disease" or "COVID-19 incidence".

Due to the large number of covariables, a selection was performed according to the criteria of correlation with statistical significance and epidemiological characteristics. Correlation between the outcome and the covariables was analyzed by Spearman's correlation coefficient. A correlation matrix was constructed to identify collinearity between the covariables. In this stage, covariables that presented significant correlation with incidence of the disease at 5% ($p < 0.05$) were selected for the modeling. In cases of correlations greater than 0.5 between the covariables, only the covariable that added the most to the linear model was included.

The outcome's spatial dependence was measured with global Moran's index (GMI). Clusters of spatial dependence were identified by calculating the Local Index of Spatial Association (LISA). We then constructed the LISA spreading maps (LISA Map) and the significance map.

## Global and local models

Global and local regression models were used to identify the best fit for COVID-19 incidence in Brazilian municipalities. The global regression models were ordinary least squares (OLS), spatial autoregressive model (SAR), and conditional autoregressive model (CAR). The OLS includes the traditional linear regression approach, taking into account the independence of observations. Diagnosis of collinearity between the covariables selected by the OLS model was verified by variance inflation factor (VIF), defining VIF values less than 10 as absence of collinearity [24]. Since the global Moran's index detected spatial dependence of COVID-19 incidence, it was necessary to control this dependence using the SAR and CAR spatial models. The SAR model incorporates the spatial dependence of the outcome (incidence of the disease). The CAR model includes the spatial effect jointly in the model's random component (error).

Among the local models, we opted to use multiscale geographically weighted regression (MGWR) to fit a regression model to each area datum, considering a bandwidth (neighborhood). The bandwidth selection method was bi-square adaptive kernel, which removes the effect of the analytical units outside the neighborhood area [25]. MGWR also employs corrected Akaike information criterion (AICc) to indicate the best size of bandwidth for each covariable. Global Moran's index was used to identify the spatial dependence of the three model's residuals.

The criteria for comparison of the final fit of the OLS, SAR, and CAR models were the $R^2$ determination coefficient and the Akaike information criterion (AIC); the former assesses the degree to which incidence of the disease can be explained by the covariables, and the latter considers the maximum likelihood and the amounts of explanatory variables used.

Preparation of the database to unify the variables, correlation analyses, and the OLS, SAR, and CAR regression models were performed in the R statistical program, version 3.6.1 [26]. The MGWR regression model's fit was assessed with the *mgwr* package in the Python programming environment [27].

## Results

A total of 4,698,163 COVID-19 cases were reported from February 25 to September 26, 2020. Spearman's correlation matrix (Table 2) shows that the covariables ELDER and PMR presented significant inverse correlation with COVID-19 incidence. The covariables BLKBRN, POP_DENS, GINI and EDUC9 showed significant positive correlation with incidence of the disease. Considering that the covariables ELDER, PPLAN, LOGINCOM, MHDI, and BLKBRN were highly correlated with each other, they were excluded from the modeling.

According to Fig 2, the highest crude incidence rates of the disease occurred in the North of Brazil and on the coastline, especially in the Northeast and Southeast.

Fig 3 shows that after Bayesian smoothing, few differences occurred in the distribution of COVID-19 incidence.

Spatial autocorrelation also corroborated this small disparity between the two distributions (crude and smoothed). Moran's index for crude incidence was 0.484 (p = 0.001), and for smoothed incidence it was 0.503 (p = 0.001). These results suggest that COVID-19 incidence presented spatial dependence between the municipalities.

Fig 4 presents the LISA Map for smoothed COVID-19 incidence. Municipalities with high incidence surrounded by neighboring municipalities with high incidence of the disease were

**Table 2. Spearman's correlation matrix.**

| | ELDER | BLKBRN | POP_DENS | DOCTRS | NURS | BEDS | ESTABS | PNEUMO | PPLAN | LOGINCOM | PMR | GINI | UNEMPL_RT | MHDI | EDUC0TO4 | EDUC5TO8 | EDUC9 | SEWAGE | GRBG | WATER |
|---|---|---|---|---|---|---|---|---|---|---|---|---|---|---|---|---|---|---|---|---|
| LOGCOV | -0.27* | 0.10* | 0.13* | 0.09* | 0.07* | 0.08* | -0.05* | -0.06* | 0.09* | 0.11* | -0.21* | 0.11* | 0.02 | 0.07* | -0.12* | -0.11* | 0.17* | -0.05* | 0.11* | 0.06* |
| ELDER | | -0.60* | -0.00 | 0.22* | 0.23* | 0.04* | 0.50* | 0.29* | 0.32* | 0.37* | 0.69* | -0.41* | -0.04* | 0.44* | -0.23* | 0.10* | 0.17* | 0.21* | 0.22* | 0.17* |
| BLKBRN | | | -0.23* | -0.37* | -0.15* | -0.05* | -0.48* | -0.21* | -0.58* | -0.73* | -0.58* | 0.51* | 0.03* | -0.73* | 0.58* | -0.17* | -0.46* | -0.29* | -0.51* | -0.27* |
| POP_DENS | | | | 0.28* | 0.01 | 0.11* | -0.02 | -0.21* | 0.38* | 0.23* | 0.10* | -0.18* | -0.04* | 0.28* | -0.34* | 0.05* | 0.33* | 0.43* | 0.42* | 0.39* |
| DOCTRS | | | | | 0.43* | 0.29* | 0.41* | 0.03* | 0.54* | 0.53* | 0.22* | -0.12* | 0.01 | 0.53* | -0.50* | 0.00 | 0.50* | 0.36* | 0.45* | 0.33* |
| NURS | | | | | | 0.26* | 0.38* | 0.07* | 0.23* | 0.26* | 0.15* | -0.04* | 0.00 | 0.29* | -0.24* | -0.07* | 0.28* | 0.15* | 0.19* | 0.19* |
| BEDS | | | | | | | 0.22* | 0.11* | 0.16* | 0.17* | 0.01 | 0.15* | 0.01 | 0.16* | -0.16* | -0.05* | 0.22* | 0.10* | 0.16* | 0.17* |
| ESTABS | | | | | | | | 0.23* | 0.37* | 0.46* | 0.40* | -0.26* | 0.01 | 0.48* | -0.36* | 0.05* | 0.32* | 0.14* | 0.28* | 0.23* |
| PNEUMO | | | | | | | | | 0.07* | 0.15* | 0.21* | -0.17* | 0.02 | 0.16* | -0.06* | 0.03* | 0.05* | -0.04* | 0.02 | -0.00 |
| PPLAN | | | | | | | | | | 0.83* | 0.34* | -0.38* | 0.06* | 0.83* | -0.76* | 0.08* | 0.75* | 0.55* | 0.75* | 0.54* |
| LOGINCOM | | | | | | | | | | | 0.39* | -0.39* | 0.04* | 0.94* | -0.86* | 0.12* | 0.80* | 0.39* | 0.73* | 0.46* |
| PMR | | | | | | | | | | | | -0.40* | -0.01 | 0.44* | -0.29* | 0.09* | 0.22* | 0.26* | 0.27* | 0.19* |
| GINI | | | | | | | | | | | | | -0.02 | -0.42* | 0.32* | -0.16* | -0.26* | -0.25* | -0.38* | -0.24* |
| UNEMPL_RT | | | | | | | | | | | | | | 0.04* | -0.01 | -0.04* | 0.05* | 0.07* | 0.05* | 0.04* |
| HDI | | | | | | | | | | | | | | | -0.89* | 0.03* | 0.89* | 0.43* | 0.75* | 0.53* |
| EDUC0TO4 | | | | | | | | | | | | | | | | -0.21* | -0.90* | -0.36* | -0.72* | -0.54* |
| EDUC5TO8 | | | | | | | | | | | | | | | | | -0.07* | -0.04* | 0.10* | 0.02 |
| EDUC9 | | | | | | | | | | | | | | | | | | 0.42* | 0.73* | 0.58* |
| SEWAGE | | | | | | | | | | | | | | | | | | | 0.55* | 0.46* |
| GRBG | | | | | | | | | | | | | | | | | | | | 0.66* |

* significant p-value

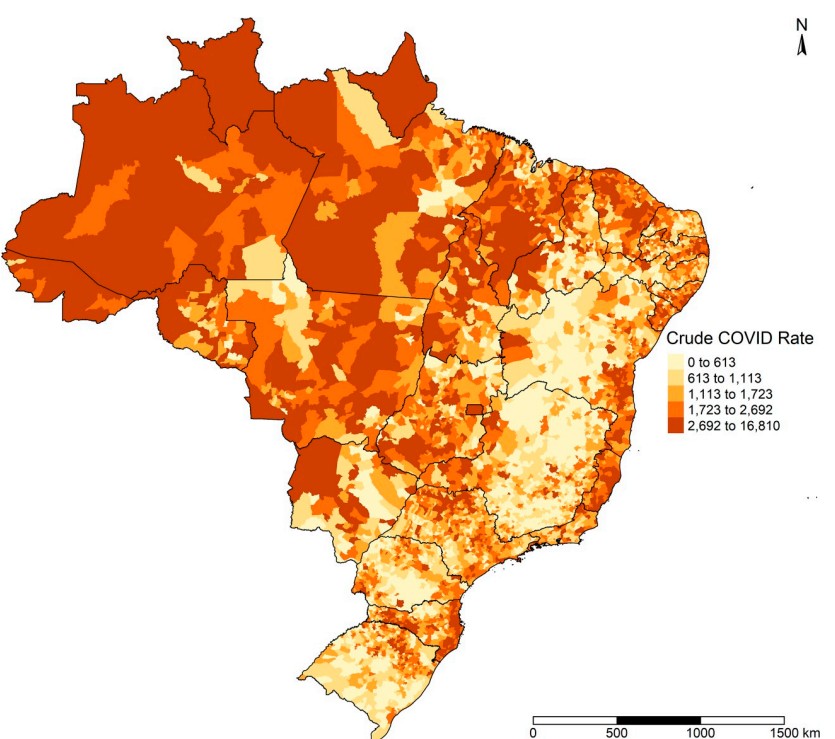

**Fig 2. Crude COVID-19 incidence in Brazilian municipalities, February 25 to September 26, 2020.**

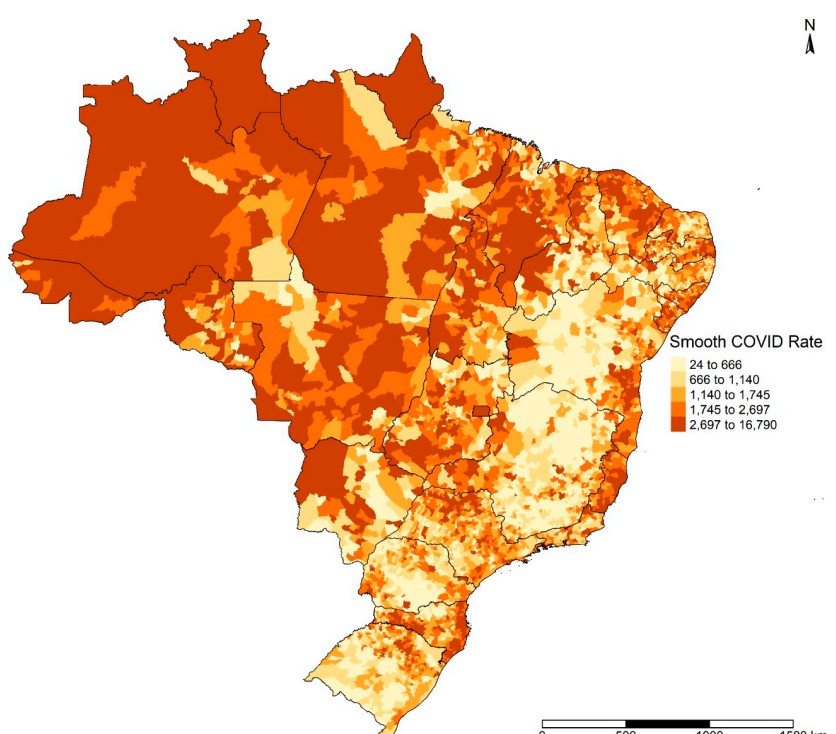

**Fig 3. Smoothed COVID-19 incidence in Brazilian municipalities, February 25 to September 26, 2020.**

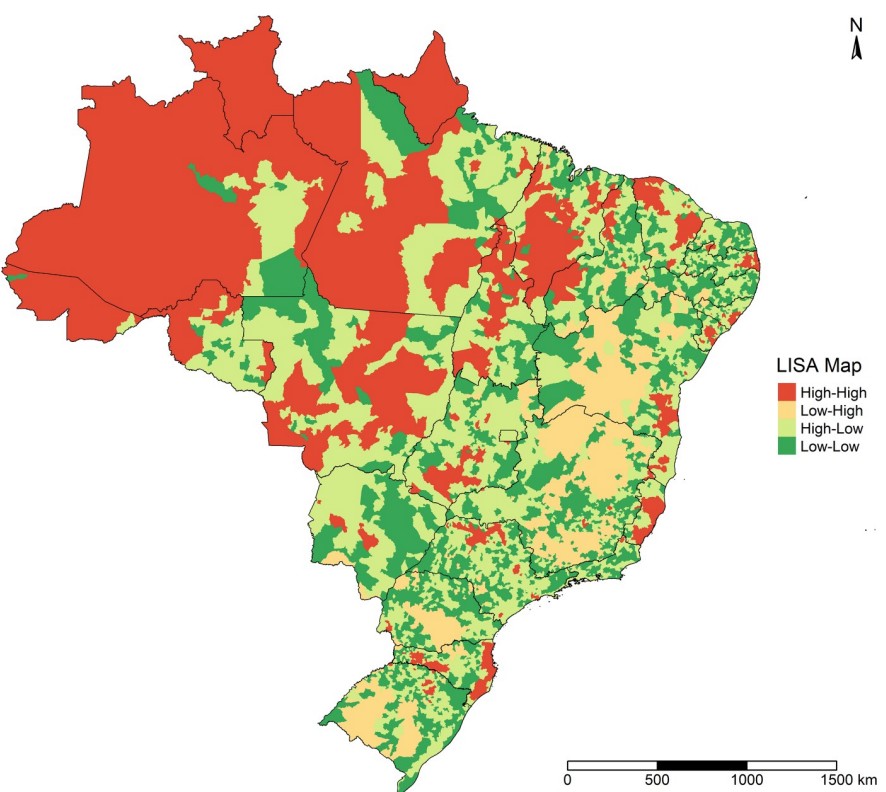

**Fig 4. LISA Map of smoothed COVID-19 incidence in Brazilian municipalities, February 25 to September 26, 2020.**

located largely in the North of Brazil; in Northeast Brazil in the state of Maranhão and on the coastline in the states of Ceará, Pernambuco, Alagoas, and Sergipe; and in Southeast Brazil in the state of Espírito Santo. Municipalities with low COVID-19 incidence surrounded by municipalities with low incidence were located mainly in the Central-West region, the interior of the Northeast and Southeast, and in much of the South.

As shown in Fig 5, the areas that presented statistical significance in the LISA Map were located mainly in a large part of the North; in Northeast Brazil, in part of the state of Maranhão, on the coastlines of the states of Ceará, Pernambuco, Alagoas, and Sergipe, and in the interior of the state of Bahia; in Southeast Brazil, in the north and south of Minas Gerais state and in Espírito Santo state; in the Central-West region, in municipalities of Mato Grosso, in the south of Tocantins and north of Goiás; in the South of Brazil, in the interior of Paraná and Rio Grande do Sul and on the coastline and in the interior of Santa Catarina.

Table 3 shows the global models' results. The higher the GINI index, the higher the incidence of the disease at the municipal level. Likewise, the higher the nurse ratio per 1,000 inhabitants in the municipalities (NURS), the higher the COVID-19 incidence. Meanwhile, proportional mortality ratio (PMR) was inversely associated with incidence of the disease. The $R^2$ and AIC of the SAR model were the best of the three models, indicating that the three covariables explained 48.9% of the variability in COVID-19 incidence in this model. The residuals from the spatial models were also controlled after adjusting for spatial dependence, since they presented GMI close to zero.

As shown in Table 4, these covariables presented VIF < 10, indicating low multicollinearity in the OLS model.

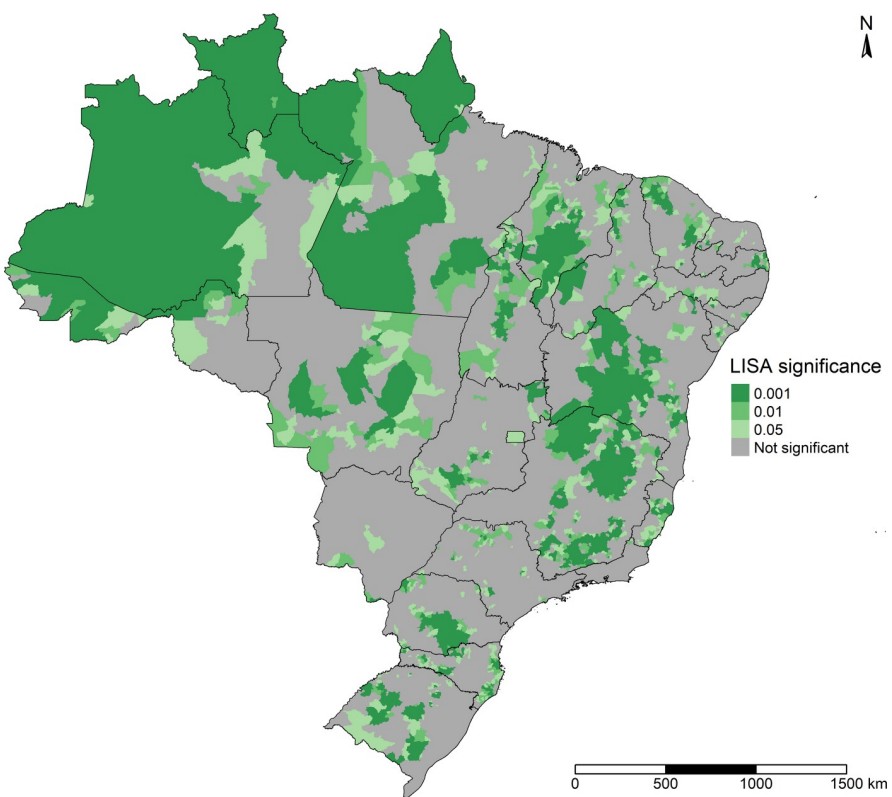

**Fig 5. Significance of the LISA Map for smoothed COVID-19 incidence in Brazilian municipalities, February 25 to September 26, 2020.**

Table 5 shows the summary results of the MGWR model. The fit improved when compared to the global regression models. $R^2$ increased to 0.699, AICc remained at 9690, and the residuals were also controlled (GMI = -0.055, p = 0.999).

The maps with the MGWR model's coefficients showed the same direction as the association found in the global regression models. The GINI index showed a significant positive association with COVID-19 incidence in half of Brazil's territory, including the entire North region, except the west of Acre State. In addition, in the parts of Maranhão, Piauí and Bahia states (Northeast region); in the Central-West region, in the north of the state of Mato Grosso; in the Southeast region, including the entire Espírito Santo and Rio de Janeiro states, as well as part of the Minas Gerais. In these areas, the GINI index was also positively associated with incidence of the disease (Fig 6).

**Table 3. Global regression models OLS, SAR, and CAR.**

|  | OLS | | SAR | | CAR | |
|---|---|---|---|---|---|---|
| **Predictors** | **Estimates** | **p-value** | **Estimates** | **p-value** | **Estimates** | **p-value** |
| **GINI** | 0.283 | 0.137 | 0.297 | 0.019 | 0.748 | <0.001 |
| **PMR** | -2.020 | <0.001 | -0.634 | <0.001 | -0.475 | <0.001 |
| **NURS** | 0.187 | <0.001 | 0.162 | <0.001 | 0.159 | <0.001 |
| **$R^2$** | 0.056 | | 0.489 | | 0.488 | |
| **AIC** | 14110 | | 10694 | | 10709 | |
| **GMI of residuals** | 0.553 | | -0.058 | | -0.061 | |

Table 4. VIF of the variables in the OLS model.

| GINI | PMR | NURS |
|---|---|---|
| 1.207 | 1.222 | 1.014 |

PMR coefficients showed a significant negative association with COVID-19 incidence in all major geographic regions of Brazil, except for a major part of the Northeast region, south of the state of Mato Grosso do Sul, west of São Paulo, north of Paraná and some municipalities of the states of Amapá, Pará and Tocantins (Fig 7).

Fig 8 illustrates the nurse ratio per 1,000 inhabitants, which showed a positive association with COVID-19 incidence throughout the North and Northeast of Brazil (except the far south of Bahia state); in Southeast Brazil, in the north, east, and south of Minas Gerais state, throughout the states of Espírito Santo and Rio de Janeiro and Greater Metropolitan São Paulo; in the Central-West region, in the north of the states of Mato Grosso and Goiás; in the South of Brazil, in the central of Santa Catarina state and part of the state of Rio Grande do Sul.

## Discussion

Spatial distribution of COVID-19 incidence was high in the North of Brazil and part of the Northeast region, including its coastline, extending to a large part of the Southeast region. Importantly, Brazil's coastline concentrates many of the country's state capitals. Multivariate analyses identified three covariables as predictors of COVID-19 incidence in Brazilian municipalities: GINI, nurse ratio per 1,000 inhabitants, and proportional mortality ratio (PMR).

Considering the covariable PMR, lower values for this indicator suggest worse levels of quality of life and economic development [28]. In the current study, proportional mortality ratio was inversely associated with COVID-19 incidence in most Brazilian municipalities (the higher the incidence, the lower the PMR), possibly because this covariable is a proxy for social development.

In the same direction, Baqui et al. [29], found higher COVID-19 mortality in the North and Northeast regions of Brazil, related to higher prevalence of comorbidities and lower socioeconomic development in these regions. Andrade et al. [11] identified high-risk clusters in six municipalities in the central-south region of Sergipe state in Northeast Brazil, including the state capital Aracaju, which has the state's highest population density and the lowest socioeconomic level. Maciel, Castro-Silva and Farias [13] found spatial dependence of COVID-19 incidence among municipalities of Ceará state in Northeast Brazil and moderate direct correlation with the municipal human development index.

High PMR values are related to low levels of social vulnerability, as in the comparison with ethnic groups. Baqui et al. [29] found higher mortality in the brown and black populations, highlighting brown ethnicity as the second leading risk factor for death, next to age. Barbosa

Table 5. Results of MGWR model.

| Predictors | Mean Z-score estimates |
|---|---|
| GINI | 0.033 |
| PMR | -0.039 |
| NURS | 0.094 |
| $R^2$ | 0.712 |
| AICc | 9,690 |
| GMI of residuals | -0.055 |

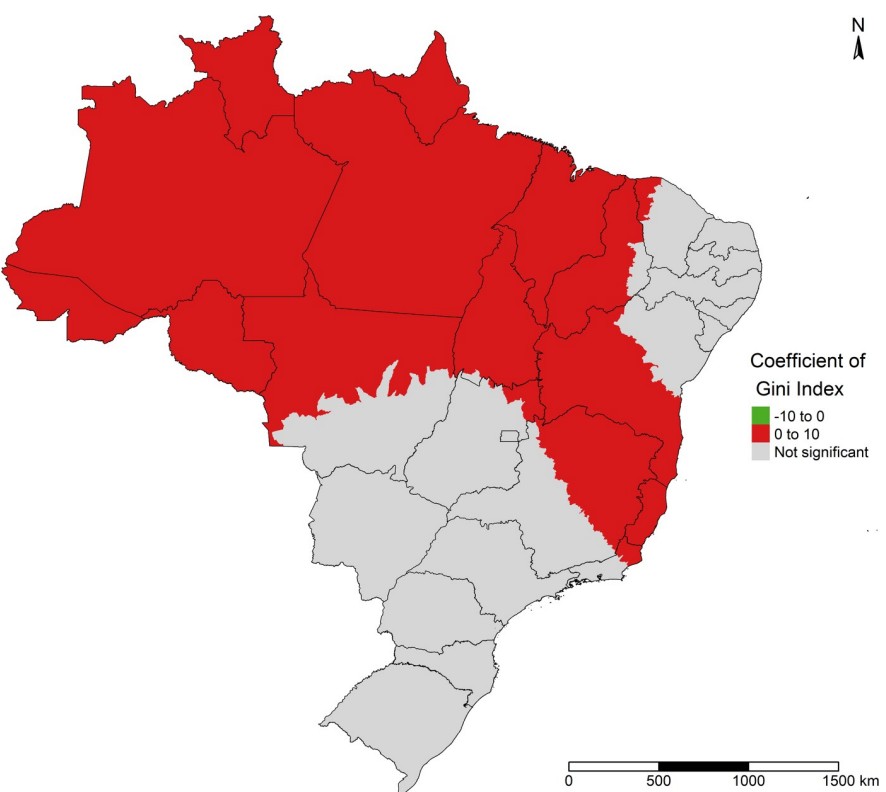

**Fig 6. Effect of the GINI index on COVID-19 incidence in Brazil.**

et al. [30] observed that incidence rate and mortality rate also revealed, respectively, a significant positive correlation with the proportion of black (Afro-Brazilian) and brown (mixed race) skinned people and with the income ratio.

Mollalo, Vahedi and Rivera [15], who evaluated global spatial models for municipalities in the United States, found directly association between COVID-19 incidence and social inequality, median household income, percentage of black female population and percentage of nurse practitioners. In the current study, the GINI index, which measures social inequality, was also directly associated with COVID-19 incidence. This association coincides with the results obtained by other studies on COVID-19 cases and deaths [18, 19] and corroborates the role of inequality as an important social determinant of health.

Cordes and Castro [14] found an inverse association between COVID-19 and higher levels of education and income in New York. However, Rafael et al. [20], in the city of Rio de Janeiro, found higher incidence in regions with high income, which suggests that access to testing occurs unequally in Brazil's reality, with the wealthiest-population having greater access to testing.

The covariable nurse ratio per 1,000 inhabitants, considered an indicator of healthcare capacity, was associated with higher COVID-19 incidence [15] and lower incidence of deaths from the disease [17]. The availability of human resources in health, represented in this study by nursing professionals, influences the capacity for detecting and reporting the disease and thus tends to increase the observed incidence.

In addition to human resources, according to a study in the United States, COVID-19 cases are underestimated, with high proportions of asymptomatic cases associated with a general lack of access to testing in a context of widespread underreporting, which can boost the search

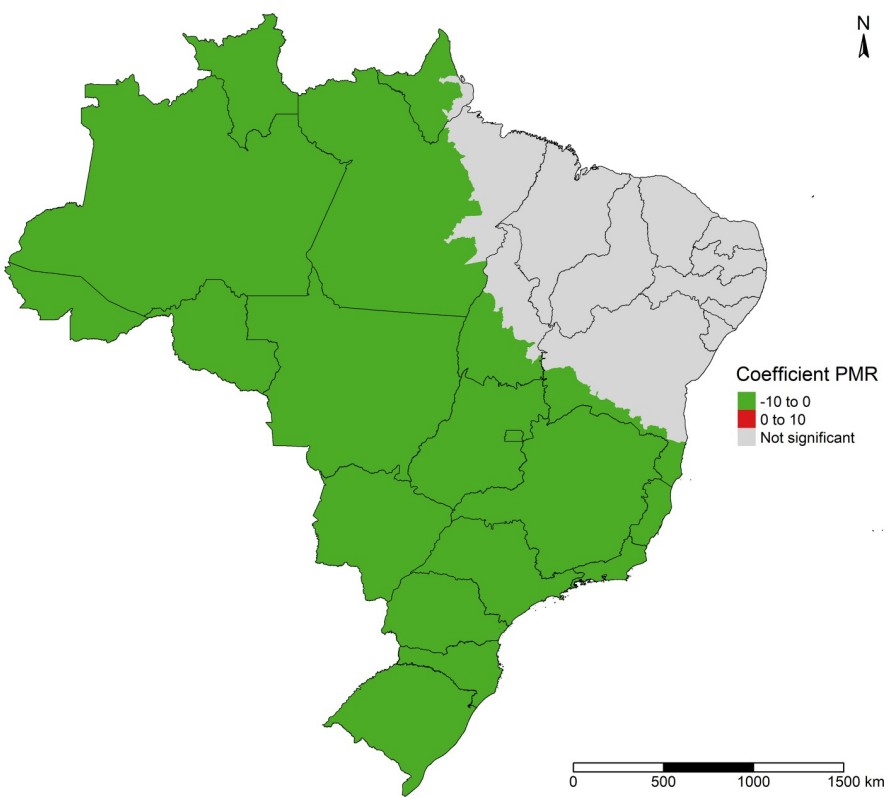

**Fig 7. Effect of proportional mortality ratio on COVID-19 incidence in Brazil.**

for healthcare services and thus explain a positive association between nursing professionals and detection of the virus [14].

The current study presents some limitations related to variations in the notification of cases between municipalities and over time, potentially introducing information bias due to under-reporting and missing data, frequent limitation in ecological studies, especially in epidemic periods. However, the inclusion of possible confounders and the use of local models allowed the analysis and identification of relevant predictors at the population level. There is also the possibility of an information bias in the covariables, since many of them were obtained from the 2010 Population Census by the IBGE, while the outcome was calculated from the number of cases reported in 2020. Some municipalities may have undergone changes in their sociodemographic characteristics, which could influence the results at the local level. However, the study assumed the non-occurrence of significant changes in the municipalities' sociodemographic profile over the course of ten years.

Although the entire population was susceptible to COVID-19 at the beginning of the pandemic, at the municipal level the disease has shown distinct repercussions, considering the various socioeconomic groups. As observed elsewhere, populations with increased social vulnerability are more exposed to the risk of infection. These populations experience limitations in adhering to social distancing measures, due to their work situation, largely informal [14], and precarious housing and sanitation conditions, factors that hinder maintenance of the hygiene needed to prevent and control transmission of the disease [13]. Effective actions to control the spread of COVID-19 and reduce the mortality from the disease should take these factors into account.

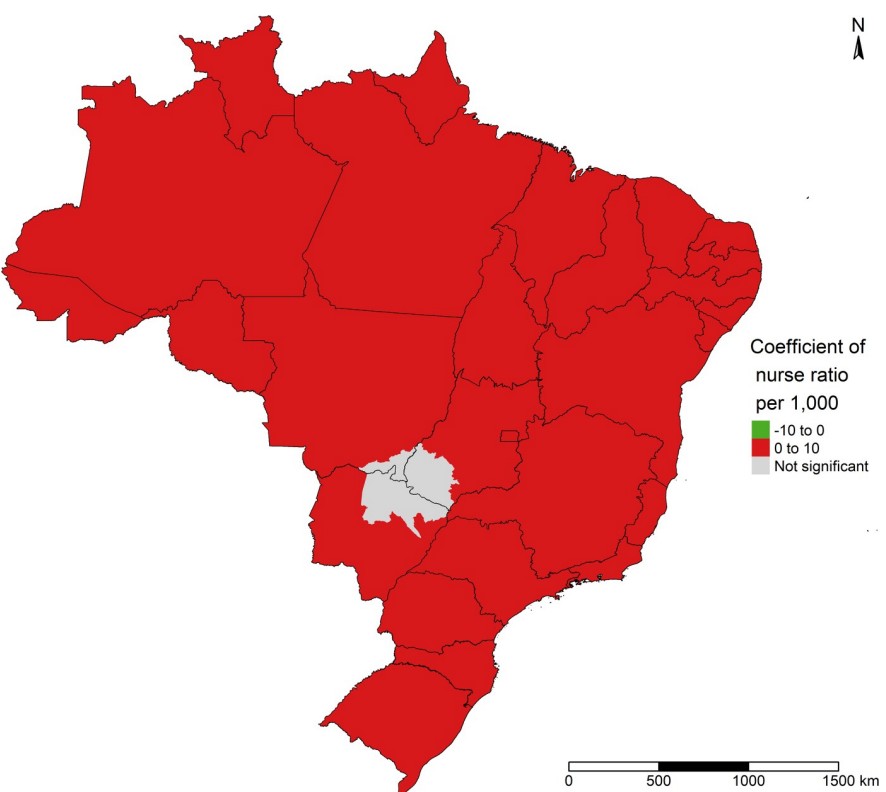

**Fig 8. Effect of nurse ratio per 1,000 inhabitants on COVID-19 incidence in Brazil.**

The findings of this research reinforced by the methodology employed helped in the global efforts to understand the spatial dynamics of COVID-19, in order to offer tools at a specific geographical level for targeted interventions.

## Author Contributions

**Conceptualization:** Carlos Eduardo Raymundo, Roberto de Andrade Medronho.

**Data curation:** Carlos Eduardo Raymundo, Marcella Cini Oliveira, Tatiana de Araujo Eleuterio, Suzana Rosa André, Marcele Gonçalves da Silva.

**Formal analysis:** Carlos Eduardo Raymundo, Marcella Cini Oliveira.

**Methodology:** Carlos Eduardo Raymundo, Roberto de Andrade Medronho.

**Supervision:** Carlos Eduardo Raymundo.

**Visualization:** Marcella Cini Oliveira, Eny Regina da Silva Queiroz, Roberto de Andrade Medronho.

**Writing – original draft:** Marcella Cini Oliveira, Tatiana de Araujo Eleuterio, Suzana Rosa André, Marcele Gonçalves da Silva, Eny Regina da Silva Queiroz, Roberto de Andrade Medronho.

**Writing – review & editing:** Carlos Eduardo Raymundo, Marcella Cini Oliveira, Tatiana de Araujo Eleuterio, Suzana Rosa André, Eny Regina da Silva Queiroz, Roberto de Andrade Medronho.

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
