## [Decision Letter · Decision Letter 0]

22 Dec 2020

PONE-D-20-35466

Spatial analysis of COVID-19 incidence and the sociodemographic context in Brazil

PLOS ONE

Dear Dr. Raymundo,

Thank you for submitting your manuscript to PLOS ONE. After careful consideration, we feel that it has merit but does not fully meet PLOS ONE’s publication criteria as it currently stands. Therefore, we invite you to submit a revised version of the manuscript that addresses the points raised during the review process.

A **rebuttal letter** that responds to **EACH** point raised by the academic editor and reviewer(s). You should upload this letter as a separate file labeled 'Response to Reviewers'.A **marked-up copy** of your manuscript that highlights changes made to the original version. You should upload this as a separate file labeled 'Revised Manuscript with Track Changes'.An **unmarked version** of your revised paper without tracked changes. You should upload this as a separate file labeled 'Manuscript'.

We look forward to receiving your revised manuscript.

Kind regards,

Brecht Devleesschauwer

Academic Editor

PLOS ONE

Journal Requirements:

2. We note that Figures 1-8 in your submission contain map images which may be copyrighted.

a. You may seek permission from the original copyright holder of Figures 1-8  to publish the content specifically under the CC BY 4.0 license. 

Reviewers' comments:

Reviewer's Responses to Questions

**Comments to the Author**

1. Is the manuscript technically sound, and do the data support the conclusions?

Reviewer #1: Yes

Reviewer #2: Yes

2. Has the statistical analysis been performed appropriately and rigorously? 

Reviewer #1: Yes

Reviewer #2: Yes

3. Have the authors made all data underlying the findings in their manuscript fully available?

Reviewer #1: Yes

Reviewer #2: Yes

4. Is the manuscript presented in an intelligible fashion and written in standard English?

Reviewer #1: Yes

Reviewer #2: Yes

5. Review Comments to the Author

Reviewer #1: COMMENTS:

1) In Methods section of the abstract, it would be fitting to mention the time period in which the data were collected in.

2) In Results section I suggest to mention the motivation why only covariates listed in Table 1 were considered. I also suggest to use more uniform spacing in Table 1.

3) Position of the legend in Figure 2 and Figure 3 in Results section needs to be adjusted so its edges don’t disrupt the map.

4) Table 2 in Results section is exceeding the page size. In general, consider reporting values rounded to two decimals.

5) I find the Discussion section a bit incoherent. For instance, on lines 269-270 the authors report the finding that lower numbers in proportional mortality ratio suggest worse levels of quality of life and economic development. On the following lines (270-278), studies are reported that suggest the contrary. I would suggest to place author’s finding into the context of studies with similar finding, in addition to the study by Baqui et al. that mentions the ethnicity effect (lines 283-286). I would also suggest to include some references for the paragraph on lines 279-282.

6) I would suggest to elaborate more on the limitations of the study. For instance, what kind of bias can be introduced ‘by the limitations related to variations in the scope, types of testing and notification of cases between municipalities and over time’ (lines 309-311)?

Reviewer #2: The manuscript aimed to analyze the spatial distribution of COVID-19 incidence in Brazil’s municipalities (counties) and investigate its association with sociodemographic determinants to better understand the social context and the epidemic’s spread in the country. The study presents the results of original research. Experiments, statistics, and other analyses are performed to a high technical standard and are described in sufficient detail.

However, I understand that a minor revision needs to be carried out in order to add quality to the manuscript.

**

Abstract:

The background does not represent the study proposal. In this section it may be interesting to better describe the spatial situation of COVID-19 in Brazil. Additionally, it is interesting to clarify what are the gaps to be filled by this investigation.

Introduction:

It is important to describe the socio-demographic measures and the inequality measures used in the studies that supported this investigation. Additionally, it is essential to explain what are the knowledge gaps and how the present manuscript intends to contribute on this topic.

Methods:

It is essential to explain the reasons for including variables in the analysis model. I understand that the variables on availability of health professionals in the analyzed territories (doctors and nurses) are another way of estimating inequalities in Brazilian municipalities. However, this option is not sufficiently explained in the manuscript.

Discussion:

Describe how the Brazilian findings can be used for other research scenarios in the world.

6. PLOS authors have the option to publish the peer review history of their article (what does this mean?). If published, this will include your full peer review and any attached files.

Reviewer #1: No

Reviewer #2: No

---

## [Author Response · Author response to Decision Letter 0]

4 Feb 2021

Dear Editor and reviewers,

Thank you very much for your comments, they were valuable to improve the paper. We included the suggested corrections in the manuscript. Below we justify our decisions about the comments and list the changes.

Please notice that the text was also modified to fulfill the requests of this journal editorial office.

Best regards.

---

## [Editor Report · Decision Letter 1]

15 Feb 2021

Spatial analysis of COVID-19 incidence and the sociodemographic context in Brazil

PONE-D-20-35466R1

Dear Dr. Raymundo,

We’re pleased to inform you that your manuscript has been judged scientifically suitable for publication and will be formally accepted for publication once it meets all outstanding technical requirements.

Kind regards,

Brecht Devleesschauwer

Academic Editor

PLOS ONE
---

## [Editor Report · Acceptance letter]

19 Feb 2021

PONE-D-20-35466R1 

Spatial analysis of COVID-19 incidence and the sociodemographic context in Brazil 

Dear Dr. Raymundo:

I'm pleased to inform you that your manuscript has been deemed suitable for publication in PLOS ONE. Congratulations! Your manuscript is now with our production department. 

Kind regards, 

on behalf of

Prof. Dr. Brecht Devleesschauwer 

Academic Editor

PLOS ONE